# Neural Expectation Maximization

**Klaus Greff**[*]
IDSIA
klaus@idsia.ch

**Sjoerd van Steenkiste**[*]
IDSIA
sjoerd@idsia.ch

**Jürgen Schmidhuber**
IDSIA
juergen@idsia.ch

## Abstract

Many real world tasks such as reasoning and physical interaction require identification and manipulation of conceptual entities. A first step towards solving these tasks is the automated discovery of distributed symbol-like representations. In this paper, we explicitly formalize this problem as inference in a spatial mixture model where each component is parametrized by a neural network. Based on the Expectation Maximization framework we then derive a differentiable clustering method that simultaneously learns how to group and represent individual entities. We evaluate our method on the (sequential) perceptual grouping task and find that it is able to accurately recover the constituent objects. We demonstrate that the learned representations are useful for next-step prediction.

## 1 Introduction

Learning useful representations is an important aspect of unsupervised learning, and one of the main open problems in machine learning. It has been argued that such representations should be distributed [13, 37] and disentangled [1, 31, 3]. The latter has recently received an increasing amount of attention, producing representations that can disentangle features like rotation and lighting [4, 12].

So far, these methods have mostly focused on the single object case whereas, for real world tasks such as reasoning and physical interaction, it is often necessary to identify and manipulate multiple entities and their relationships. In current systems this is difficult, since superimposing multiple distributed and disentangled representations can lead to ambiguities. This is known as the Binding Problem [21, 37, 13] and has been extensively discussed in neuroscience [33]. One solution to this problem involves learning a separate representation for each object. In order to allow these representations to be processed identically they must be described in terms of the same (disentangled) features. This would then avoid the binding problem, and facilitate a wide range of tasks that require knowledge about individual objects. This solution requires a process known as *perceptual grouping*: dynamically splitting (segmenting) each input into its constituent conceptual entities.

In this work, we tackle this problem of learning how to group and efficiently represent individual entities, in an unsupervised manner, based solely on the statistical structure of the data. Our work follows a similar approach as the recently proposed Tagger [7] and aims to further develop the understanding, as well as build a theoretical framework, for the problem of *symbol-like representation learning*. We formalize this problem as inference in a spatial mixture model where each component is parametrized by a neural network. Based on the Expectation Maximization framework we then derive a differentiable clustering method, which we call *Neural Expectation Maximization* (N-EM). It can be trained in an unsupervised manner to perform perceptual grouping in order to learn an efficient representation for each group, and naturally extends to sequential data.

---

[*]Both authors contributed equally to this work.

## 2 Neural Expectation Maximization

The goal of training a system that produces separate representations for the individual conceptual entities contained in a given input (here: image) depends on what notion of *entity* we use. Since we are interested in the case of unsupervised learning, this notion *can only* rely on statistical properties of the data. We therefore adopt the intuitive notion of a conceptual entity as being a common cause (the object) for multiple observations (the pixels that depict the object). This common cause induces a dependency-structure among the affected pixels, while the pixels that correspond to different entities remain (largely) independent. Intuitively this means that knowledge about some pixels of an object helps in predicting its remainder, whereas it does not improve the predictions for pixels of other objects. This is especially obvious for sequential data, where pixels belonging to a certain object share a common fate (e.g. move in the same direction), which makes this setting particularly appealing.

We are interested in representing each entity (object) $k$ with some vector $\boldsymbol{\theta}_k$ that captures all the structure of the affected pixels, but carries no information about the remainder of the image. This modularity is a powerful invariant, since it allows the same representation to be reused in different contexts, which enables generalization to novel combinations of known objects. Further, having all possible objects represented in the same format makes it easier to work with these representations. Finally, having a separate $\boldsymbol{\theta}_k$ for each object (as opposed to for the entire image) allows $\boldsymbol{\theta}_k$ to be distributed and disentangled without suffering from the binding problem.

We treat each image as a composition of $K$ objects, where each pixel is determined by exactly one object. Which objects are present, as well as the corresponding assignment of pixels, varies from input to input. Assuming that we have access to the family of distributions $P(\boldsymbol{x}|\boldsymbol{\theta}_k)$ that corresponds to an object level representation as described above, we can model each image as a mixture model. Then Expectation Maximization (EM) can be used to simultaneously compute a Maximum Likelihood Estimate (MLE) for the individual $\boldsymbol{\theta}_k$-s and the grouping that we are interested in.

The central problem we consider in this work is therefore how to learn such a $P(\boldsymbol{x}|\boldsymbol{\theta}_k)$ in a completely unsupervised fashion. We accomplish this by parametrizing this family of distributions by a differentiable function $f_\phi(\boldsymbol{\theta})$ (a neural network with weights $\phi$). We show that in that case, the corresponding EM procedure becomes fully differentiable, which allows us to backpropagate an appropriate outer loss into the weights of the neural network. In the remainder of this section we formalize and derive this method which we call *Neural Expectation Maximization* (N-EM).

### 2.1 Parametrized Spatial Mixture Model

We model each image $\boldsymbol{x} \in \mathbb{R}^D$ as a spatial mixture of $K$ components parametrized by vectors $\boldsymbol{\theta}_1, \ldots, \boldsymbol{\theta}_K \in \mathbb{R}^M$. A differentiable non-linear function $f_\phi$ (a neural network) is used to transform these representations $\boldsymbol{\theta}_k$ into parameters $\psi_{i,k} = f_\phi(\boldsymbol{\theta}_k)_i$ for separate pixel-wise distributions. These distributions are typically Bernoulli or Gaussian, in which case $\psi_{i,k}$ would be a single probability or a mean and variance respectively. This parametrization assumes that given the representation, the pixels are independent but *not* identically distributed (unlike in standard mixture models). A set of binary latent variables $\boldsymbol{Z} \in [0,1]^{D \times K}$ encodes the unknown true pixel assignments, such that $z_{i,k} = 1$ iff pixel $i$ was generated by component $k$, and $\sum_k z_{i,k} = 1$. A graphical representation of this model can be seen in Figure 1, where $\boldsymbol{\pi} = (\pi_1, \ldots \pi_K)$ are the mixing coefficients (or prior for $\boldsymbol{z}$). The full likelihood for $\boldsymbol{x}$ given $\boldsymbol{\theta} = (\boldsymbol{\theta}_1, \ldots, \boldsymbol{\theta}_K)$ is given by:

$$P(\boldsymbol{x}|\boldsymbol{\theta}) = \prod_{i=1}^{D} \sum_{\boldsymbol{z}_i} P(x_i, \boldsymbol{z}_i|\boldsymbol{\psi}_i) = \prod_{i=1}^{D} \sum_{k=1}^{K} \underbrace{P(z_{i,k} = 1)}_{\pi_k} P(x_i|\psi_{i,k}, z_{i,k} = 1). \tag{1}$$

### 2.2 Expectation Maximization

Directly optimizing $\log P(\boldsymbol{x}|\boldsymbol{\psi})$ with respect to $\boldsymbol{\theta}$ is difficult due to marginalization over $\mathbf{z}$, while for many distributions optimizing $\log P(\boldsymbol{x}, \mathbf{z}|\boldsymbol{\psi})$ is much easier. Expectation Maximization (EM; [6]) takes advantage of this and instead optimizes a lower bound given by the expected log likelihood:

$$\mathcal{Q}(\boldsymbol{\theta}, \boldsymbol{\theta}^{\text{old}}) = \sum_{\mathbf{z}} P(\mathbf{z}|\boldsymbol{x}, \boldsymbol{\psi}^{\text{old}}) \log P(\boldsymbol{x}, \mathbf{z}|\boldsymbol{\psi}). \tag{2}$$

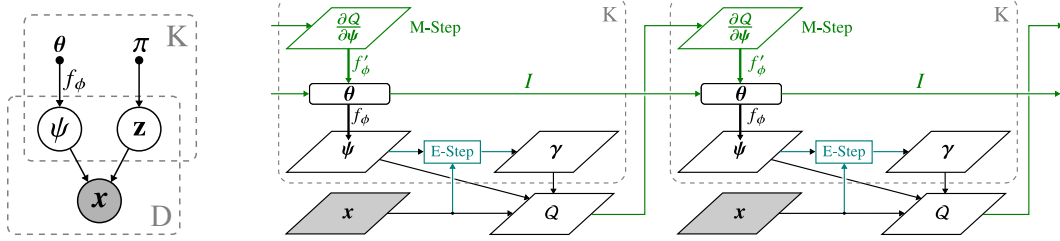

Figure 1: **left:** The probabilistic graphical model that underlies N-EM. **right:** Illustration of the computations for two steps of N-EM.

Iterative optimization of this bound alternates between two steps: in the *E-step* we compute a new estimate of the posterior probability distribution over the latent variables given $\theta^{\text{old}}$ from the previous iteration, yielding a new soft-assignment of the pixels to the components (clusters):

$$\gamma_{i,k} := P(z_{i,k} = 1 | x_i, \psi_i^{\text{old}}). \tag{3}$$

In the M-step we then aim to find the configuration of $\theta$ that would maximize the expected log-likelihood using the posteriors computed in the E-step. Due to the non-linearity of $f_\phi$ there exists no analytical solution to $\arg\max_\theta \mathcal{Q}(\theta, \theta^{\text{old}})$. However, since $f_\phi$ is differentiable, we can improve $\mathcal{Q}(\theta, \theta^{\text{old}})$ by taking a gradient ascent step:[2]

$$\theta^{\text{new}} = \theta^{\text{old}} + \eta \frac{\partial \mathcal{Q}}{\partial \theta} \qquad \text{where} \qquad \frac{\partial \mathcal{Q}}{\partial \theta_k} \propto \sum_{i=1}^{D} \gamma_{i,k}(\psi_{i,k} - x_i)\frac{\partial \psi_{i,k}}{\partial \theta_k}. \tag{4}$$

The resulting algorithm belongs to the class of *generalized EM algorithms* and is guaranteed (for a sufficiently small learning rate $\eta$) to converge to a (local) optimum of the data log likelihood [42].

## 2.3   Unrolling

In our model the information about statistical regularities required for clustering the pixels into objects is encoded in the neural network $f_\phi$ with weights $\phi$. So far we have considered $f_\phi$ to be fixed and have shown how we can compute an MLE for $\theta$ alongside the appropriate clustering. We now observe that by unrolling the iterations of the presented generalized EM, we obtain an end-to-end differentiable clustering procedure based on the statistical model implemented by $f_\phi$. We can therefore use (stochastic) gradient descent and fit the statistical model to capture the regularities corresponding to objects for a given dataset. This is implemented by back-propagating an appropriate loss (see Section 2.4) through "time" (BPTT; [39, 41]) into the weights $\phi$. We refer to this trainable procedure as *Neural Expectation Maximization* (N-EM), an overview of which can be seen in Figure 1.

Upon inspection of the structure of N-EM we find that it resembles $K$ copies of a recurrent neural network with hidden states $\theta_k$ that, at each timestep, receive $\gamma_k \odot (\psi_k - x)$ as their input. Each copy generates a new $\psi_k$, which is then used by the E-step to re-estimate the soft-assignments $\gamma$. In order to accurately mimic the M-Step (4) with an RNN, we must impose several restrictions on its weights and structure: the "encoder" must correspond to the Jacobian $\partial \psi_k/\partial \theta_k$, and the recurrent update must linearly combine the output of the encoder with $\theta_k$ from the previous timestep. Instead, we introduce a new algorithm named RNN-EM, when substituting that part of the computational graph of N-EM with an actual RNN (without imposing any restrictions). Although RNN-EM can no longer guarantee

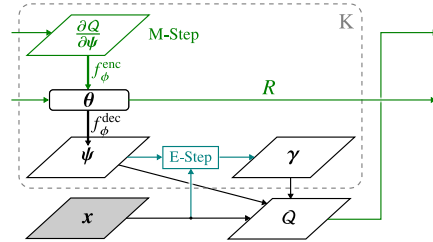

Figure 2: RNN-EM Illustration. Note the changed encoder and recurrence compared to Figure 1.

convergence of the data log likelihood, its recurrent weights increase the flexibility of the clustering procedure. Moreover, by using a fully parametrized recurrent weight matrix RNN-EM naturally extends to sequential data. Figure 2 presents the computational graph of a single RNN-EM timestep.

## 2.4 Training Objective

N-EM is a differentiable clustering procedure, whose outcome relies on the statistical model $f_\phi$. We are interested in a particular unsupervised clustering that corresponds to grouping entities based on the statistical regularities in the data. To train our system, we therefore require a loss function that teaches $f_\phi$ to map from representations $\theta$ to parameters $\psi$ that correspond to pixelwise distributions for such objects. We accomplish this with a two-term loss function that guides each of the $K$ networks to model the structure of a single object independently of any other information in the image:

$$L(\boldsymbol{x}) = - \sum_{i=1}^{D} \sum_{k=1}^{K} \underbrace{\gamma_{i,k} \log P(x_i, z_{i,k} | \psi_{i,k})}_{\text{intra-cluster loss}} - \underbrace{(1 - \gamma_{i,k}) D_{KL}[P(x_i) || P(x_i | \psi_{i,k}, z_{i,k})]}_{\text{inter-cluster loss}}. \quad (5)$$

The intra-cluster loss corresponds to the same expected data log-likelihood $\mathcal{Q}$ as is optimized by N-EM. It is analogous to a standard reconstruction loss used for training autoencoders, weighted by the cluster assignment. Similar to autoencoders, this objective is prone to trivial solutions in case of overcapacity, which prevent the network from modelling the statistical regularities that we are interested in. Standard techniques can be used to overcome this problem, such as making $\theta$ a bottleneck or using a noisy version of $x$ to compute the inputs to the network. Furthermore, when RNN-EM is used on sequential data we can use a next-step prediction loss.

Weighing the loss pixelwise is crucial, since it allows each network to specialize its predictions to an individual object. However, it also introduces a problem: the loss for out-of-cluster pixels ($\gamma_{i,k} = 0$) vanishes. This leaves the network free to predict anything and does not yield specialized representations. Therefore, we add a second term (inter-cluster loss) which penalizes the KL divergence between out-of-cluster predictions and the pixelwise prior of the data. Intuitively this tells each representation $\theta_k$ to contain no information regarding non-assigned pixels $x_i$: $P(x_i | \psi_{i,k}, z_{i,k}) = P(x_i)$.

A disadvantage of the interaction between $\gamma$ and $\psi$ in (5) is that it may yield conflicting gradients. For any $\theta_k$ the loss for a given pixel $i$ can be reduced by better predicting $x_i$, or by decreasing $\gamma_{i,k}$ (i.e. taking less responsibility) which is (due to the E-step) realized by being worse at predicting $x_i$. A practical solution to this problem is obtained by stopping the $\gamma$ gradients, i.e. by setting $\partial L / \partial \gamma = 0$ during backpropagation.

## 3 Related work

The method most closely related to our approach is Tagger [7], which similarly learns perceptual grouping in an unsupervised fashion using $K$ copies of a neural network that work together by reconstructing different parts of the input. Unlike in case of N-EM, these copies additionally learn to output the grouping, which gives Tagger more direct control over the segmentation and supports its use on complex texture segmentation tasks. Our work maintains a close connection to EM and relies on the posterior inference of the E-Step as a grouping mechanism. This facilitates theoretical analysis and simplifies the task for the resulting networks, which we find can be markedly smaller than in Tagger. Furthermore, Tagger does not include any recurrent connections on the level of the hidden states, precluding it from next step prediction on sequential tasks.[3]

The Binding problem was first considered in the context of Neuroscience [21, 37] and has sparked some early work in oscillatory neural networks that use synchronization as a grouping mechanism [36, 38, 24]. Later, complex valued activations have been used to replace the explicit simulation of oscillation [25, 26]. By virtue of being general computers, any RNN can in principle learn a suitable mechanism. In practice however it seems hard to learn, and adding a suitable mechanism like competition [40], fast weights [29], or perceptual grouping as in N-EM seems necessary.

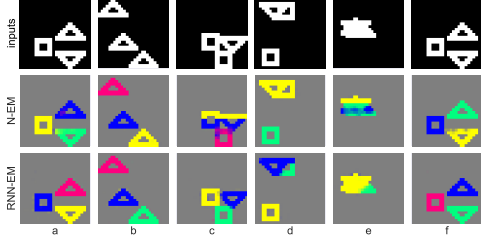

Figure 3: Groupings by RNN-EM (bottom row), N-EM (middle row) for six input images (top row). Both methods recover the individual shapes accurately when they are separated (a, b, f), even when confronted with the same shape (b). RNN-EM is able to handle most occlusion (c, d) but sometimes fails (e). The exact assignments are permutation invariant and depend on $\gamma$ initialization; compare (a) and (f).

Unsupervised Segmentation has been studied in several different contexts [30], from random vectors [14] over texture segmentation [10] to images [18, 16]. Early work in unsupervised video segmentation [17] used generalized Expectation Maximization (EM) to infer how to split frames of moving sprites. More recently optical flow has been used to train convolutional networks to do figure/ground segmentation [23, 34]. A related line of work under the term of multi-causal modelling [28] has formalized perceptual grouping as inference in a generative compositional model of images. Masked RBMs [20] for example extend Restricted Boltzmann Machines with a latent mask inferred through Block-Gibbs sampling.

Gradient backpropagation through inference updates has previously been addressed in the context of sparse coding with (Fast) Iterative Shrinkage/Tresholding Algorithms ((F)ISTA; [5, 27, 2]). Here the unrolled graph of a fixed number of ISTA iterations is replaced by a recurrent neural network that parametrizes the gradient computations and is trained to predict the sparse codes directly [9]. We derive RNN-EM from N-EM in a similar fashion and likewise obtain a trainable procedure that has the structure of iterative pursuit built into the architecture, while leaving tunable degrees of freedom that can improve their modeling capabilities [32]. An alternative to further empower the network by untying its weights across iterations [11] was not considered for flexibility reasons.

## 4 Experiments

We evaluate our approach on a perceptual grouping task for generated static images and video. By composing images out of simple shapes we have control over the statistical structure of the data, as well as access to the ground-truth clustering. This allows us to verify that the proposed method indeed recovers the intended grouping and learns representations corresponding to these objects. In particular we are interested in studying the role of next-step prediction as a unsupervised objective for perceptual grouping, the effect of the hyperparameter $K$, and the usefulness of the learned representations.

In all experiments we train the networks using ADAM [19] with default parameters, a batch size of 64 and $50\,000$ train + $10\,000$ validation + $10\,000$ test inputs. Consistent with earlier work [8, 7], we evaluate the quality of the learned groupings with respect to the ground truth while ignoring the background and overlap regions. This comparison is done using the Adjusted Mutual Information (AMI; [35]) score, which provides a measure of clustering similarity between 0 (random) and 1 (perfect match). We use early stopping when the validation loss has not improved for 10 epochs.[4] A detailed overview of the experimental setup can be found in Appendix A. All reported results are averages computed over five runs.[5]

### 4.1 Static Shapes

To validate that our approach yields the intended behavior we consider a simple perceptual grouping task that involves grouping three randomly chosen regular shapes ($\triangle\triangledown\square$) located in random positions of $28 \times 28$ binary images [26]. This simple setup serves as a test-bed for comparing N-EM and RNN-EM, before moving on to more complex scenarios.

We implement $f_\phi$ by means of a single layer fully connected neural network with a sigmoid output $\psi_{i,k}$ for each pixel that corresponds to the mean of a Bernoulli distribution. The representation $\boldsymbol{\theta}_k$ is

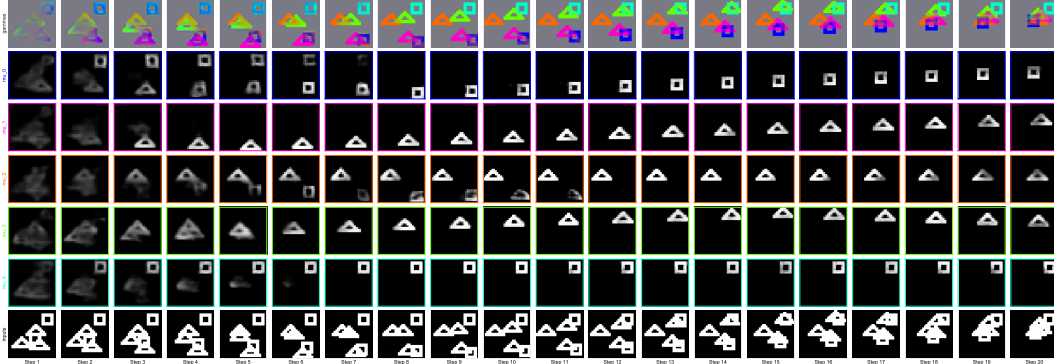

Figure 4: A sequence of 5 shapes flying along random trajectories (bottom row). The next-step prediction of each copy of the network (rows 2 to 5) and the soft-assignment of the pixels to each of the copies (top row). Observe that the network learns to separate the individual shapes as a means to efficiently solve next-step prediction. Even when many of the shapes are overlapping, as can be seen in time-steps 18-20, the network is still able to disentangle the individual shapes from the clutter.

a real-valued 250-dimensional vector squashed to the $(0, 1)$ range by a sigmoid function before being fed into the network. Similarly for RNN-EM we use a recurrent neural network with 250 sigmoidal hidden units and an equivalent output layer. Both networks are trained with $K = 3$ and unrolled for 15 EM steps.

As shown in Figure 3, we observe that both approaches are able to recover the individual shapes as long as they are separated, even when confronted with identical shapes. N-EM performs worse if the image contains occlusion, and we find that RNN-EM is in general more stable and produces considerably better groupings. This observation is in line with findings for Sparse Coding [9]. Similarly we conclude that the tunable degrees of freedom in RNN-EM help speed-up the optimization process resulting in a more powerful approach that requires fewer iterations. The benefit is reflected in the large score difference between the two: $0.826 \pm 0.005$ AMI compared to $0.475 \pm 0.043$ AMI for N-EM. In comparison, Tagger achieves an AMI score of $0.79 \pm 0.034$ (and $0.97 \pm 0.009$ with layernorm), while using about twenty times more parameters [7].

## 4.2 Flying Shapes

We consider a sequential extension of the *static shapes* dataset in which the shapes ($\triangle \triangledown \square$) are floating along random trajectories and bounce off walls. An example sequence with 5 shapes can be seen in the bottom row of Figure 4. We use a convolutional encoder and decoder inspired by the discriminator and generator networks of infoGAN [4], with a recurrent neural network of 100 sigmoidal units (for details see Section A.2). At each timestep $t$ the network receives $\boldsymbol{\gamma}_k(\boldsymbol{\psi}_k^{(t-1)} - \tilde{\boldsymbol{x}}^{(t)})$ as input, where $\tilde{\boldsymbol{x}}^{(t)}$ is the current frame corrupted with additional bitflip noise ($p = 0.2$). The next-step prediction objective is implemented by replacing $x$ with $x^{(t+1)}$ in (5), and is evaluated at each time-step.

Table 1 summarizes the results on flying shapes, and an example of a sequence with 5 shapes when using $K = 5$ can be seen in Figure 4. For 3 shapes we observe that the produced groupings are close to perfect (AMI: $0.970 \pm 0.005$). Even in the very cluttered case of 5 shapes the network is able to separate the individual objects in almost all cases (AMI: $0.878 \pm 0.003$).

These results demonstrate the adequacy of the next step prediction task for perceptual grouping. However, we find that the converse also holds: the corresponding representations are useful for the prediction task. In Figure 5 we compare the next-step prediction error of RNN-EM with $K = 1$ (which reduces to a recurrent autoencoder that receives the difference between its previous prediction and the current frame as input) to RNN-EM with $K = 5$ on this task. To evaluate RNN-EM on next-step prediction we computed its loss using $P(x_i|\boldsymbol{\psi}_i) = P(x_i|\max_k \psi_{i,k})$ as opposed to $P(x_i|\boldsymbol{\psi}_i) = \sum_k \gamma_{i,k} P(x_i|\psi_{i,k})$ to avoid including information from the next timestep. The reported BCE loss for RNN-EM is therefore an *upperbound* to the true BCE loss. From the figure we observe that RNN-EM produces significantly lower errors, especially when the number of objects increases.

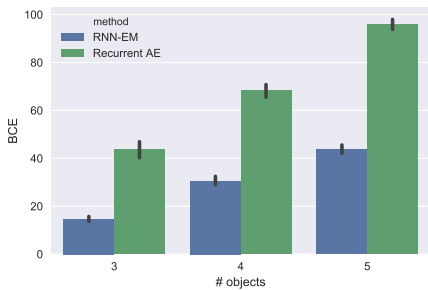

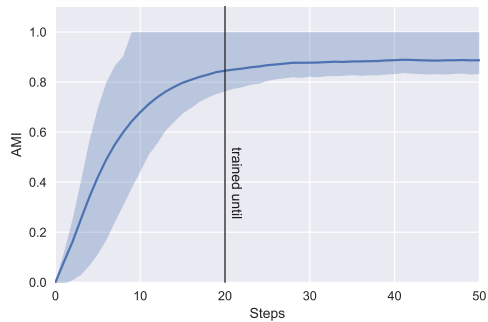

Figure 5: Binomial Cross Entropy Error obtained by RNN-EM and a recurrent autoencoder (RNN-EM with $K = 1$) on the denoising and next-step prediction task. RNN-EM produces significantly lower BCE across different numbers of objects.

Figure 6: Average AMI score (blue line) measured for RNN-EM (trained for 20 steps) across the flying MNIST test-set and corresponding quartiles (shaded areas), computed for each of 50 time-steps. The learned grouping dynamics generalize to longer sequences and even further improve the AMI score.

| Train | | | Test | | | Test Generalization | | |
|---|---|---|---|---|---|---|---|---|
| # obj. | K | AMI | # obj. | K | AMI | # obj. | K | AMI |
| 3 | 3 | $0.969 \pm 0.006$ | 3 | 3 | $0.970 \pm 0.005$ | 3 | 5 | $0.972 \pm 0.007$ |
| 3 | 5 | $0.997 \pm 0.001$ | 3 | 5 | $0.997 \pm 0.002$ | 3 | 3 | $0.914 \pm 0.015$ |
| 5 | 3 | $0.614 \pm 0.003$ | 5 | 3 | $0.614 \pm 0.003$ | 3 | 3 | $0.886 \pm 0.010$ |
| 5 | 5 | $0.878 \pm 0.003$ | 5 | 5 | $0.878 \pm 0.003$ | 3 | 5 | $0.981 \pm 0.003$ |

Table 1: AMI scores obtained by RNN-EM on *flying shapes* when varying the number of objects and number of components $K$, during training and at test time.

Finally, in Table 1 we also provide insight about the impact of choosing the hyper-parameter $K$, which is unknown for many real-world scenarios. Surprisingly we observe that training with too large $K$ is in fact favourable, and that the network learns to leave the excess groups empty. When training with too few components we find that the network still learns about the individual shapes and we observe only a slight drop in score when correctly setting the number of components at test time. We conclude that RNN-EM is robust towards different choices of $K$, and specifically that choosing $K$ to be too high is not detrimental.

### 4.3 Flying MNIST

In order to incorporate greater variability among the objects we consider a sequential extension of MNIST. Here each sequence consists of gray-scale $24 \times 24$ images containing two down-sampled MNIST digits that start in random positions and float along randomly sampled trajectories within the image for $T$ timesteps. An example sequence can be seen in the bottom row of Figure 7.

We deploy a slightly deeper version of the architecture used in flying shapes. Its details can be found in Appendix A.3. Since the images are gray-scale we now use a Gaussian distribution for each pixel with fixed $\sigma^2 = 0.25$ and $\mu = \psi_{i,k}$ as computed by each copy of the network. The training procedure is identical to flying shapes except that we replace bitflip noise with masked uniform noise: we first sample a binary mask from a multi-variate Bernoulli distribution with $p = 0.2$ and then use this mask to interpolate between the original image and samples from a Uniform distribution between the minimum and maximum values of the data (0,1).

We train with $K = 2$ and $T = 20$ on flying MNIST having two digits and obtain an AMI score of $0.819 \pm 0.022$ on the test set, measured across 5 runs.

In early experiments we observed that, given the large variability among the 50 000 unique digits, we can boost the model performance by training in stages using 20, 500, 50 000 digits. Here we exploit the generalization capabilities of RNN-EM to quickly transfer knowledge from a less varying set of

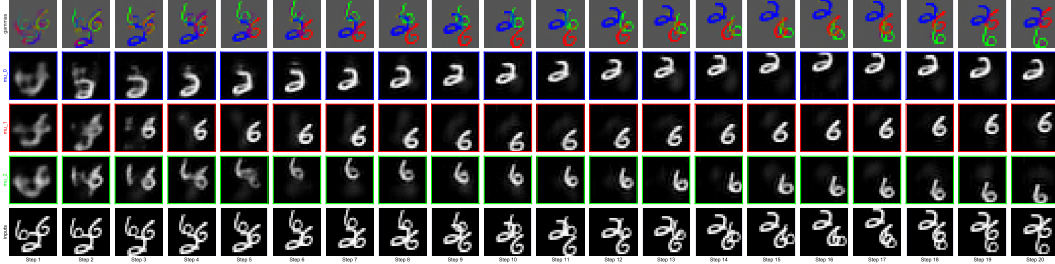

Figure 7: A sequence of 3 MNIST digits flying across random trajectories in the image (bottom row). The next-step prediction of each copy of the network (rows 2 to 4) and the soft-assignment of the pixels to each of the copies (top row). Although the network was trained (stage-wise) on sequences with two digits, it is accurately able to separate three digits.

MNIST digits to unseen variations. We used the same hyper-parameter configuration as before and obtain an AMI score of $0.917 \pm 0.005$ on the test set, measured across 5 runs.

We study the generalization capabilities and robustness of these trained RNN-EM networks by means of three experiments. In the first experiment we evaluate them on flying MNIST having three digits (one extra) and likewise set $K = 3$. Even without further training we are able to maintain a high AMI score of $0.729 \pm 0.019$ (stage-wise: $0.838 \pm 0.008$) on the test-set. A test example can be seen in Figure 7. In the second experiment we are interested in whether the grouping mechanism that has been learned can be transferred to static images. We find that using 50 RNN-EM steps we are able to transfer a large part of the learned grouping dynamics and obtain an AMI score of $0.619 \pm 0.023$ (stage-wise: $0.772 \pm 0.008$) for two static digits. As a final experiment we evaluate the directly trained network on the same dataset for a larger number of timesteps. Figure 6 displays the average AMI score across the test set as well as the range of the upper and lower quartile for each timestep.

The results of these experiments confirm our earlier observations for flying shapes, in that the learned grouping dynamics are robust and generalize across a wide range of variations. Moreover we find that the AMI score further improves at test time when increasing the sequence length.

## 5 Discussion

The experimental results indicate that the proposed Neural Expectation Maximization framework can indeed learn how to group pixels according to constituent objects. In doing so the network learns a useful and localized representation for individual entities, which encodes only the information relevant to it. Each entity is represented separately in the same space, which avoids the binding problem and makes the representations usable as efficient symbols for arbitrary entities in the dataset. We believe that this is useful for reasoning in particular, and a potentially wide range of other tasks that depend on interaction between multiple entities. Empirically we find that the learned representations are already beneficial in next-step prediction with multiple objects, a task in which overlapping objects are problematic for standard approaches, but can be handled efficiently when learning a separate representation for each object.

As is typical in clustering methods, in N-EM there is no preferred assignment of objects to groups and so the grouping numbering is arbitrary and only depends on initialization. This property renders our results permutation invariant and naturally allows for *instance segmentation*, as opposed to semantic segmentation where groups correspond to pre-defined categories. RNN-EM learns to segment in an unsupervised fashion, which makes it applicable to settings with little or no labeled data. On the downside this lack of supervision means that the resulting segmentation may not always match the intended outcome. This problem is inherent to this task since in real world images the notion of an object is ill-defined and task dependent. We envision future work to alleviate this by extending unsupervised segmentation to hierarchical groupings, and by dynamically conditioning them on the task at hand using top-down feedback and attention.

# 6 Conclusion

We have argued for the importance of separately representing conceptual entities contained in the input, and suggested clustering based on statistical regularities as an appropriate unsupervised approach for separating them. We formalized this notion and derived a novel framework that combines neural networks and generalized EM into a trainable clustering algorithm. We have shown how this method can be trained in a fully unsupervised fashion to segment its inputs into entities, and to represent them individually. Using synthetic images and video, we have empirically verified that our method can recover the objects underlying the data, and represent them in a useful way. We believe that this work will help to develop a theoretical foundation for understanding this important problem of unsupervised learning, as well as providing a first step towards building practical solutions that make use of these symbol-like representations.

## Acknowledgements

The authors wish to thank Paulo Rauber and the anonymous reviewers for their constructive feedback. This research was supported by the Swiss National Science Foundation grant 200021_165675/1 and the EU project "INPUT" (H2020-ICT-2015 grant no. 687795). We are grateful to NVIDIA Corporation for donating us a DGX-1 as part of the Pioneers of AI Research award, and to IBM for donating a "Minsky" machine.

## Footnotes

[2]Here we assume that $P(x_i | z_{i,k} = 1, \psi_{i,k})$ is given by $\mathcal{N}(x_i; \mu = \psi_{i,k}, \sigma^2)$ for some fixed $\sigma^2$, yet a similar update arises for many typical parametrizations of pixel distributions.

[3]RTagger [15]: a recurrent extension of Tagger that does support sequential data was developed concurrent to this work.

[4]Note that we do not stop on the AMI score as this is not part of our objective function and only measured to evaluate the performance *after* training.

[5]Code to reproduce all experiments is available at `https://github.com/sjoerdvansteenkiste/Neural-EM`

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
