[Supplementary Material]

# A  Experiment Details

The following subsections provide detailed information about the experimental setup of our empirical evaluation.

In all experiments we train the networks using ADAM [19] with default parameters, a batch size of 64 and $50\,000$ train + $10\,000$ validation + $10\,000$ test inputs. The quality of the learned groupings is evaluated by computing the Adjusted Mutual Information (AMI; [35]) with respect to the ground truth, while ignoring the background and overlap regions (as is consistent with earlier work [8, 7]). We use early stopping when the validation loss has not improved for 10 epochs.

## A.1  Experiments on Static Shapes

Each input consists of a $28 \times 28$ binary image containing three regular shapes ($\triangle\triangledown\square$) located in random positions [26].

For N-EM we implement $f_\phi$ by means of a single layer fully connected neural network with a sigmoid activation function. It receives a real-valued 250-dimensional vector $\boldsymbol{\theta}$ as input and outputs for each pixel a value that parameterizes a Bernoulli distribution. We squash $\boldsymbol{\theta}$ with a Sigmoid before passing it to the network and train an additional weight to implement the learning rate that is used to combine the gradient ascent updates into the current parameter estimate.

Similarly for RNN-EM we use a recurrent neural network with 250 Sigmoidal hidden units and an fully-connected output-layer with a sigmoid activation function that parametrizes a Bernoulli distribution for each pixel in the same fashion.

We train both networks with $K = 4$ for 15 EM steps and add bitflip noise with probability 0.1 to each of the pixels. The prior for each pixel in the data is set to a Bernoulli distribution with $p = 0$. The outer-loss is only injected at the final EM-step.

## A.2  Experiments on Flying Shapes

Each input consists of a sequence of binary $28 \times 28$ images containing a fixed number of shapes ($\triangle\triangledown\square$) that start in random positions and float along randomly sampled trajectories within the image for 20 steps.

We use a convolutional encoder-decoder architecture inspired by recent GANs [4] with a recurrent neural network as bottleneck:

1. $4 \times 4$ conv. 32 ELU. stride 2. layer norm

2. $4 \times 4$ conv. 64 ELU. stride 2. layer norm

3. fully connected. 512 ELU. layer norm

4. recurrent. 100 Sigmoid. layer norm on the output

5. fully connected. 512 RELU. layer norm

6. fully connected. $7 \times 7 \times 64$ RELU. layer norm

7. $4 \times 4$ reshape 2 nearest-neighbour, conv. 32 RELU. layer norm

8. $4 \times 4$ reshape 2 nearest-neighbour, conv. 1 Sigmoid

Instead of using transposed convolutions (to implement the "de-convolution") we first reshape the image using the default nearest-neighbour interpolation followed by a normal convolution in order to avoid frequency artifacts [22]. Note that we do not add layer norm on the recurrent connection.

At each timestep $t$ we feed $\boldsymbol{\gamma}_k(\boldsymbol{\psi}_k^{(t-1)} - \tilde{\boldsymbol{x}}^{(t)})$ as input to the network, where $\tilde{\boldsymbol{x}}$ is the input with added bitflip noise ($p = 0.2$). RNN-EM is trained with a next-step prediction objective implemented by replacing $\boldsymbol{x}$ with $\boldsymbol{x}^{(t+1)}$ in (5), which we evaluate at each time-step. A single RNN-EM step is used for each timestep. The prior for each pixel in the data is set to a Bernoulli distribution with $p = 0$. We prevent conflicting gradient updates by not back-propagating any gradients through $\boldsymbol{\gamma}$.

### A.3 Experiments on Flying MNIST

Each input consists of a sequence of gray-scale $24 \times 24$ images containing a fixed number of down-sampled (by a factor of two along each dimension) MNIST digits that start in random positions and "fly" across randomly sampled trajectories within the image for $T$ timesteps.

We use a slightly deeper version of the architecture used for flying shapes:

1. $4 \times 4$ conv. 32 ELU. stride 2. layer norm
2. $4 \times 4$ conv. 64 ELU. stride 2. layer norm
3. $4 \times 4$ conv. 128 ELU. stride 2. layer norm
4. fully connected. 512 ELU. layer norm
5. recurrent. 250 Sigmoid. layer norm on the output
6. fully connected. 512 RELU. layer norm
7. fully connected. $3 \times 3 \times 128$ RELU. layer norm
8. $4 \times 4$ reshape 2 nearest-neighbour, conv. 64 RELU. layer norm
9. $4 \times 4$ reshape 2 nearest-neighbour, conv. 32 RELU. layer norm
10. $4 \times 4$ reshape 2 nearest-neighbour, conv. 1 linear

The training procedure is largely identical to the one described for flying shapes except that we replace the bitflip noise with masked uniform noise: we first sample a binary mask from a multi-variate Bernoulli distribution with $p = 0.2$ and then use this mask to interpolate between the original image and samples from a Uniform distribution between the minimum $(0.0)$ and maximum $(1.0)$ values of the data. We use a learning rate of $0.0005$ (from the second stage onwards in case of stage-wise training), scale the second-loss term by a factor of $0.2$ and find it beneficial to normalize the masked differences between the prediction and the image (zero mean, standard deviation one) before passing it to the network.