[Reviews · NeurIPS 2017]

Reviewer 1



In this work author propose a new framework that combines neural networks and the EM algorithm to derive an approach to grouping and learning individual entities. The proposed approach is evaluated on a perceptual grouping task for generated images and videos. The described empirical study on the three data types shows the potential of the approach. My only concern is there isn’t really a solid baseline to compare against. In a summary I think that this is an interesting work. I should also note that It seems that a shorter version of this paper was published on the ICLR 2017 workshop track. Figure 6 should be properly labeled.

Reviewer 2



This paper presents some though-provoking experiments in unsupervised entity recognition from time-series data. For me the impact of the paper came in Figs 3 and 5, which showed a very human-like decomposition. I'm not convinced that analyzing a few static shapes is an important problem these days. To me, it seems like a "first step" toward a more significant problem of recognizing concurrent actions (In this case, they have actions like "flying triangle" and "flying 9", with occasional occlusions muddying the picture). For example, RNN-EM running on non-pixel input features (output from a static object detector output (YOLO?)) seems one reasonable comparison point. - line 109: B[space]presents - 135: "stopping the gamma gradients" (and something similar in Supplemental). Not exactly sure what this means. Is it setting those gradients to zero? or simply not backpropagating at all through those layers? - 215: Figure 4.2 should be Figure 4? - 209 and 220: Table 4.2 should be Table 1? If training with higher K is favorable, why use low K for flying MNIST? In Table 1, what happens at test time if you fail in "correctly setting the number of components at test time? Do you get empty clusters if K > # objects? I guess I'd like some perspective about the perennial problem of choosing the "correct" number of clusters at test time. For flying MNIST, carefully stepwise training seems complicated and not transferable to more general cases. For me, this seems like a disadvantage. My hope would be that the non-digit [line segment, line 239] "unsupervised" statistically relevant groupings could still be used as input features to train a net whose output matches the human concept of complete number shapes. What happens if you see new objects at test time, or if the number of object classes gets very large? Recognizing line segments or other moving subfeatures again no longer be such a bad idea or disadvantage. I would have enjoyed seeing them tackle approaches with grouping together "flying [this collection of statistical features]".

Reviewer 3



The paper presents two related deep neural networks for clustering entities together into roughly independant components. In their implementations the entities are always pixels in an image. The first technique they brand as neural expectation maximization, and show how the network is performing an unrolled form of generalized EM. The E-step predicts the probability that a given pixel was generated by each of the components. The M-step, then updates the parameters of each component using these probabilities. This is neural, because each of these steps relies on a neural network which maps from the parameters of a component to the pixel values. By unrolling this process they can train this network in an end-to-end basis. They also proposed a related model which modifies this process by updating the parameters of the components at each step in the unrolling using an RNN-style update. This allows them to extend the model to sequential/video data where the image is changing at each step. The show results only on toy datasets consisting of a small number of shapes or MNIST images (3-5) superimposed on top of each other. Their neural EM approach performs significantly worse than recent work on Tagger for static data, however the RNN version of their model slightly out performs Tagger. However Tagger significantly outperforms even the RNN version of their work if Tagger is trained with batch normalization. In the video setting, their model performs significantly better, which is not surprising given the additional information available in video. They show both higher scores for predicting the pixel to component mapping, but also much better reconstruction MSE than a standard (fully entangled) autoencoder model (which is the same a their model with 1 component). Overall these seems like a solid paper with solid results on a toy dataset. My main concern is that the technique may not work on messier real world data, since they only show results on relatively toy datasets where the components are completely independant, and the component boundaries are relatively clear when they are not overlapping. Results in a more realistic domain would make the paper much strong, but even in the current state I believe both the technique and the results are interesting.